# Dynamics of the Gut Microbiome in *Shigella*-Infected Children during the First Two Years of Life

Esther Ndungo,[a,b] Johanna B. Holm,[c,d] Syze Gama,[e] Andrea G. Buchwald,[a,b] Sharon M. Tennant,[a,f] Miriam K. Laufer,[a,b] Marcela F. Pasetti,[a,b] David A. Rasko[c,d]

aCenter for Vaccine Development and Global Health, University of Maryland School of Medicine, Baltimore, Maryland, USA
bDepartment of Pediatrics, University of Maryland School of Medicine, Baltimore, Maryland, USA
cInstitute for Genome Sciences, University of Maryland School of Medicine, Baltimore, Maryland, USA
dDepartment of Microbiology and Immunology, University of Maryland School of Medicine, Baltimore, Maryland, USA
eBlantyre Malaria Project, University of Malawi College of Medicine, Blantyre, Malawi
fDepartment of Medicine, University of Maryland School of Medicine, Baltimore, Maryland, USA

**ABSTRACT** *Shigella* continues to be a major contributor to diarrheal illness and dysentery in children younger than 5 years of age in low- and middle-income countries. Strategies for the prevention of shigellosis have focused on enhancing adaptive immunity. The interaction between *Shigella* and intrinsic host factors, such as the microbiome, remains unknown. We hypothesized that *Shigella* infection would impact the developing microbial community in infancy and, conversely, that changes in the gastrointestinal microbiome may predispose infections. To test this hypothesis, we characterized the gastrointestinal microbiota in a longitudinal birth cohort from Malawi that was monitored for *Shigella* infection using 16S rRNA amplicon sequencing. Children with at least one *Shigella* quantitative polymerase chain reaction (qPCR) positive sample during the first 2 years of life (cases) were compared to uninfected controls that were matched for sex and age. Overall, the microbial species diversity, as measured by the Shannon diversity index, increased over time, regardless of case status. At early time points, the microbial community was dominated by *Bifidobacterium longum* and *Escherichia/Shigella*. A greater abundance of *Prevotella* 9 and *Bifidobacterium kashiwanohense* was observed at 2 years of age. While no single species was associated with susceptibility to *Shigella* infection, significant increases in *Lachnospiraceae* NK4A136 and *Fusicatenibacter saccharivorans* were observed following *Shigella* infection. Both taxa are in the family Lachnospiraceae, which are known short-chain fatty acid producers that may improve gut health. Our findings identified temporal changes in the gastrointestinal microbiota associated with *Shigella* infection in Malawian children and highlight the need to further elucidate the microbial communities associated with disease susceptibility and resolution.

**IMPORTANCE** *Shigella* causes more than 180 million cases of diarrhea globally, mostly in children living in poor regions. Infection can lead to severe health impairments that reduce quality of life. There is increasing evidence that disruptions in the gut microbiome early in life can influence susceptibility to illnesses. A delayed or impaired reconstitution of the microbiota following infection can further impact overall health. Aiming to improve our understanding of the interaction between *Shigella* and the developing infant microbiome, we investigated changes in the gut microbiome of *Shigella*-infected and uninfected children over the course of their first 2 years of life. We identified species that may be involved in recovery from *Shigella* infection and in driving the microbiota back to homeostasis. These findings support future studies into the elucidation of the interaction between the microbiota and enteric pathogens in young children and into the identification of potential targets for prevention or treatment.

**KEYWORDS** *Shigella*, gut microbiome, infant microbiome

Address correspondence to Marcela F. Pasetti, mpasetti@som.umaryland.edu, or David A. Rasko, drasko@som.umaryland.edu.

The authors declare no conflict of interest.

*S*higella is one of the top three causes of moderate to severe diarrhea (MSD) in the first 5 years of life in children living in Asia and Sub-Saharan Africa (1–4). It is an invasive enteric pathogen that causes mucosal inflammation and the disruption of the intestinal barrier (5, 6), leading to watery diarrhea and dysentery (bloody diarrhea) (4). Frequent and repeated bouts of diarrheal disease in children result in debilitating sequelae, including impaired growth and stunting, deficits in cognitive development, and an increased risk of metabolic syndrome (7–9), which can result in lifelong health impairments (10, 11). The transmissibility and clinical severity of disease in this vulnerable group, along with the emergence of antibiotic-resistant strains, make *Shigella* prevention a public health priority. While efforts to determine the roles of innate and adaptive immunity in preventing and reducing susceptibility to *Shigella* infection in infants are ongoing (12–17), the interaction between *Shigella* and the gastrointestinal microbiome, as well as its subsequent consequences to the gastrointestinal and the overall host health, remain largely unexplored.

The colonization of the gastrointestinal tract in infants is dynamic and sensitive to many factors, such as gestational age (18–20), mode of delivery (vaginal or cesarean delivery) (18, 21), and nutritional status (e.g., breastfeeding status and malnutrition) (22–26), and these can vary greatly across different geographical settings (27, 28). There has been an increased interest in understanding the role of the gastrointestinal microbiota in promoting health, including its potential to reduce illnesses early in life (29–31). Conversely, disruptions in the gut microbiome early in life have been associated with gastrointestinal disorders, such as necrotizing enterocolitis (NEC), inflammatory bowel disease (IBD) (30), and infectious diarrhea (32, 33). However, the roles of commensal bacteria (presence or absence), either acting as a barrier to *Shigella* infection or facilitating its colonization, and the influence of *Shigella* infection in establishing the developing microbiota in early life have not been characterized.

Previous studies exploring the role of the microbiota in infectious disease in children have generally focused on diarrhea (32–34) or on other pathogens, (e.g., cholera [35–37] or diarrheagenic *E. coli* infection [38]). An epidemiological study on the impact of *Shigella* infection on the infant microbiome relied on data obtained from a single time point (cross-sectional samples, [39]), and therefore failed to capture the dynamic nature of the microbiome preceding and succeeding *Shigella* infection. We hypothesized that *Shigella* infection would impact the evolving microbial community in infancy and, conversely, that changes in the gastrointestinal microbiome may predispose future infections. To address this question, we conducted a longitudinal characterization of the gastrointestinal microbiota in children from birth to 2 years of age living in Malawi, where *Shigella* is endemic and is a major contributor to diarrheal disease (40). Using 16S rRNA amplicon sequencing, we examined the relationship between microbial communities and the incidence of *Shigella* infection. The dynamic composition of the microbiome postinfection was also examined. We identified temporal changes in the microbiota and an increased abundance of taxa consistent with health recovery following infection. The study highlights the dynamic nature of the microbiome in early life and the impact of *Shigella* infection on the developing microbiome.

## RESULTS

**Cohort and sample summary.** Participants were recruited as part of a mother-infant cohort in a malaria surveillance study (41). All of the infants were born between February and November of 2016. Rectal swab samples were collected every 6 months during routine well-child visits, as well as each time the child presented at the clinic with diarrhea. Of the 369 rectal swabs collected, 37 (10%) were positive for *Shigella* by (qPCR), which matches the overall *Shigella* prevalence over the first 2 years of life in children from 8 countries enrolled in the MAL-ED study (42). Most of the *Shigella* qPCR positive samples were detected in children older than 12 months of age (25 out of 37, 69%) (Fig. 1A), consistent with the results of previous studies (3, 43). In addition, most of the *Shigella* positive samples (29 out of 37, 81%) were collected between the months of November and April, which coincides with the rainy season in Malawi (Fig. 1A) (44), reflecting the previously described

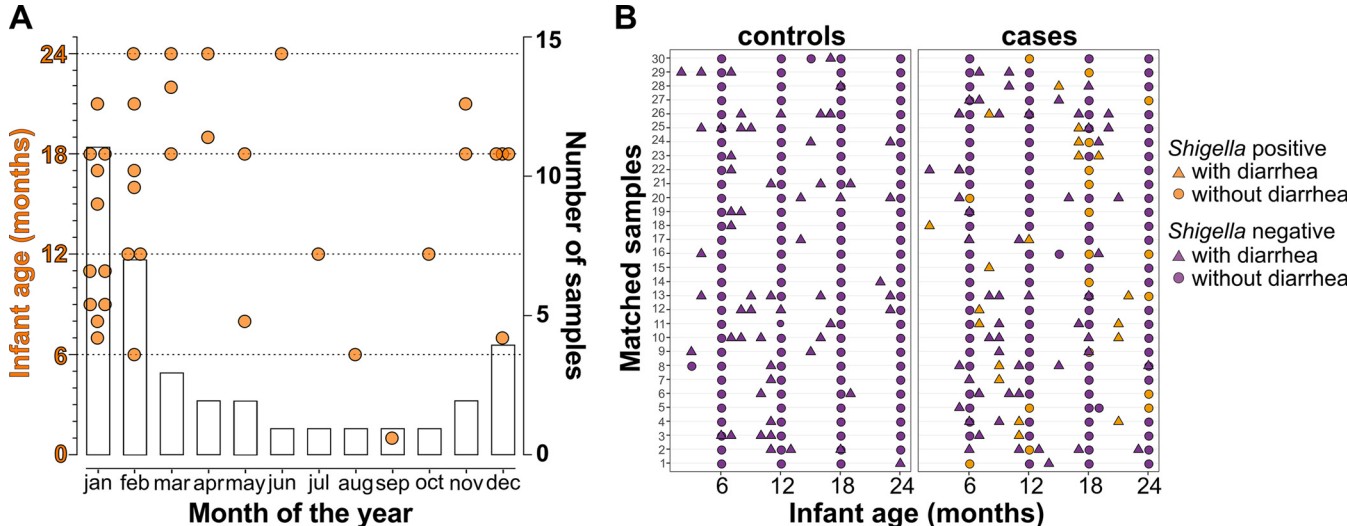

**FIG 1** Cohort sample summary. (A) Number of samples collected (right *y* axis) and infant age at sample collection (left *y* axis) in each month of the year (*x* axis). The orange symbols represent samples that were *Shigella* quantitative polymerase chain reaction (qPCR) positive. (B) Sample matching strategy: samples from matched cases and controls from 0 to 24 months of age. Each row represents samples collected at each time point from each matched pair of cases (right panel) and controls (left panel). The triangles represent samples collected with diarrhea, and the circles represent samples with no diarrhea. The orange symbols represent samples that were *Shigella* qPCR positive.

seasonality of *Shigella* infections (42). The earliest time point of a rectal swab collection was 2 months of age.

Out of 90 children whose rectal swabs were tested for *Shigella* by qPCR, 33 (37%) had at least one *Shigella* qPCR positive sample during the 24-month study period. Of these, 30 children with a complete set of longitudinal swab samples from each of the 6-, 12-, 18-, and 24-month visits were included in our analysis and classified as cases. To reduce the effects of age and sex as potential confounders, 30 children (with no *Shigella* qPCR positive samples) matching the cases by sex and age (born within the same month or as close as possible), were included as matched controls (Fig. 1B). We also noted whether samples were collected on "diarrheal" visits (as defined by the health care provider). Diarrheal samples that were negative by *Shigella* qPCR were deemed to have diarrhea caused by a pathogen other than *Shigella* (Fig. 1B).

Both the case and the control groups had similar weights at birth (means of 3.2 kg and 3.1 kg, respectively), and there were no significant differences between the ages of the mothers (means of 27 years and 28 years, respectively) or the gestational ages (means of 38 and 39 weeks, respectively). Each group of children was composed of 50% males and females. Length for age *z* scores (LAZ), a measure for stunting which has been associated with *Shigella* infection in children (9, 45–48), did not significantly differ between the case and control groups at birth or at 24 months of age (Table 1).

The total read counts did not differ significantly between the *Shigella* positive samples and the negative samples (Fig. S1A), nor did the *Escherichia/Shigella* read counts (Fig. S1B). However, for the *Shigella* qPCR positive samples, the mean *Escherichia/Shigella* read count in the diarrheal samples was significantly higher than that of the samples without diarrhea (Fig. S1C). For the *Shigella* qPCR positive samples, we did not observe an association between qPCR quantitation cycle (Cq) values and *Escherichia/Shigella* read counts (Fig. S1D).

**Longitudinal variation in microbiome composition.** We first set out to identify temporal changes in the gastrointestinal microbiota by calculating the alpha diversity as measured by the Shannon diversity index (SDI). SDI was positively correlated with age ($R^2 = 0.25$) and increased during the first 2 years for all infants (Fig. 2A). The increase in SDI did not differ between the cases and the controls ($F = 2.78$, $P = 0.096$) (Fig. 2B).

For an overview of the taxonomic composition of the gastrointestinal microbiomes during the first 2 years of life, we focused on the top 10 identified taxa based on the relative

**TABLE 1** Cohort characteristics

| Characteristics | Cases[a] (n = 30) | Controls[b] (n = 30) | P value[c] |
|---|---|---|---|
| Female Sex, No. (%) | 15 (50%) | 15 (50%) | |
| Birthweight, mean in kg (range) | 3.2 (1.5 to 4.3) | 3.1 (2.0 to 4.3) | 0.69 |
| Length-for-age z-scores | | | |
| at birth, mean (range) | 0.04 (−6.25 to 6.40) | −0.21 (−1.96 to 2.60) | 0.62 |
| at 24 mo, mean (range) | −1.71 (−3.54 to −0.09) | −1.74 (−4.37 to 1.44) | 0.88 |
| Maternal age at birth, mean no. of years (range) | 27 (17 to 38) | 28 (17 to 45) | 0.36 |
| Gestational age at birth, mean no. of weeks (range) | 38 (33 to 45) | 39 (34 to 44) | 0.10 |

[a]Cases: infants with at least one *Shigella* quantitative polymerase chain reaction (qPCR) positive sample.
[b]Controls: infants with no *Shigella* qPCR positive samples.
[c]P values are from unpaired *t* tests comparing the cases and the controls.

abundance for all individuals in the cohort (Fig. 3A; Table S1). The infant gastrointestinal microbiota was enriched with species from the phyla Actinobacteria, Bacteriodetes, Firmicutes, and Proteobacteria. These included *Bifidobacterium* spp., which have been previously demonstrated to be enriched in the gastrointestinal microbiome of breastfed infants (49, 50), *Prevotella* spp., which are commonly isolated at higher frequencies in fecal samples isolated from African populations (32, 39, 51), as well as *Escherichia/Shigella*, (16S rRNA sequencing does not distinguish these separately), which would be expected in *Shigella* positive samples and are common in infant gastrointestinal microbiomes (52, 53).

We identified shifts in the abundance of *Bifidobacterium longum*, which dominated the microbial community at early time points but decreased significantly ($P < 0.05$) by 24 months (13.3% of the sequenced reads at 6 months versus 0.7% at 24 months) (Fig. 3A and B; Table S1). Of interest, another member of the bifidobacterial species, *Bifidobacterium kashiwanohense* exhibited the opposite trend, increasing in relative abundance over time (0.3% of the sequenced reads at 6 months compared to 3.5% of the sequenced reads at 24 months). The mean relative abundance of the *Escherichia/Shigella* group decreased significantly from 4.4% of the sequencing reads at 6 months to 0.9% of the sequencing reads at 18 months ($P < 0.05$). The opposite was observed for *Prevotella* 9, which had a relative abundance of 2.4% of the sequenced reads at 6 months but dominated the microbiota at month 24 (9.7% of the sequenced reads). The mean relative abundance of each of the top 10 taxa at months 6, 12, 18, and 24 is shown in Fig. 3A and B and in Table S1.

**Differences in microbiota dynamics in cases and controls at the time of, prior to, and following *Shigella* infection.** To identify changes in the gastrointestinal microbiota that are concomitant with *Shigella* infection, we compared the alpha diversity at the first *Shigella* qPCR positive visit in the cases (the index visit) and their matched controls. After adjusting for variables that could affect alpha diversity, including infant age

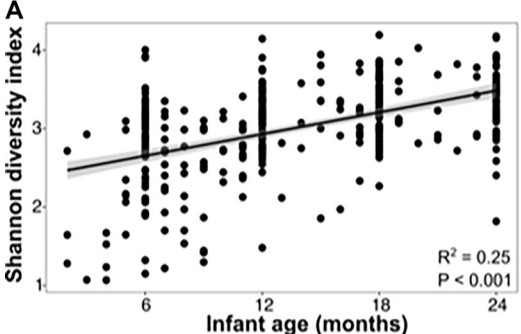 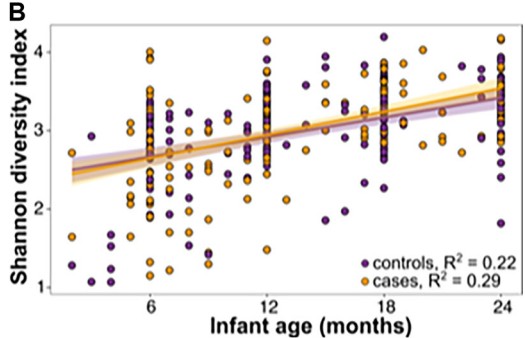

**FIG 2** Alpha diversity comparisons in infant microbiomes. (A) Shannon diversity indices at different ages at the time of sample collection, from 2 months to 24 months of age. The symbols represent individual values. The $R^2$ and $P$ values are from a simple linear regression model. (B) Shannon diversity indices of samples from case (orange) versus control (purple) individuals at the time of sample collection. The symbols represent individual values. The $R^2$ values from simple linear regression models are shown separately for the cases and the controls within the graph.

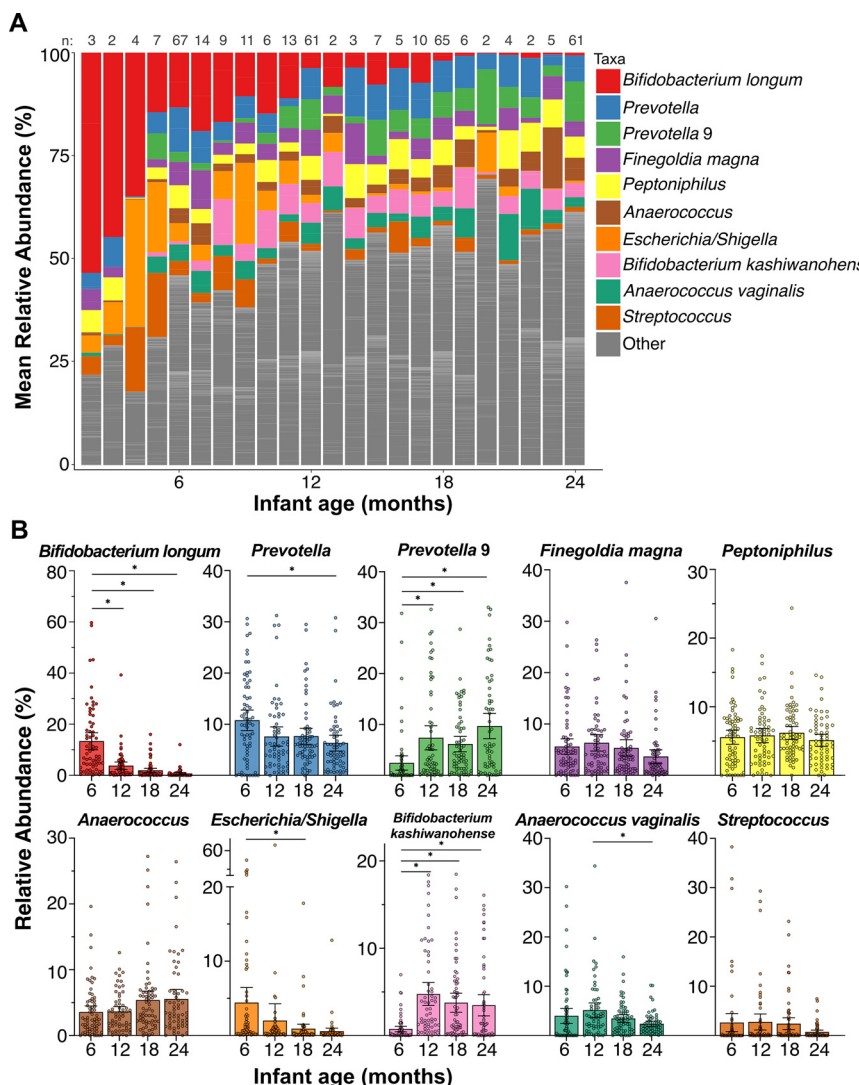

**FIG 3** Taxonomic compositions of infant gastrointestinal microbiomes by age. (A) Mean relative abundance of the 10 most abundant taxa by infant age. n represents the number of samples at each time point (month). (B) Bar graphs showing the mean relative abundances (%) of the top 10 most abundant taxa at the infant ages of 6, 12, 18, and 24 months. The symbols represent individual values. The mean relative abundances at different ages were compared via a one-way analysis of variance (ANOVA). *, $P < 0.05$.

(32, 33, 38), diarrhea (32, 54), and antibiotic use (55, 56), we determined that the SDI did not differ significantly by *Shigella* infection status ($P = 0.52$) (Table S2). Notably, alpha diversity was significantly associated with infant age ($P < 0.001$), diarrhea ($P = 0.012$), and recent antibiotic use ($P = 0.017$) (Table S2).

Considering that the presence or absence of certain microbial community members may facilitate or inhibit *Shigella* infection, we examined whether any bacterial communities were differentially abundant between the cases and the matched controls at the index visit. While *Escherichia/Shigella* were among the taxa increased in the cases, no taxa were significantly associated with case status (*Shigella* infected versus controls) after controlling for multiple variables (Fig. S2).

We then examined changes in the gastrointestinal microbiota that potentially predisposed infants to *Shigella* infection. Considering all samples immediately prior to the case or control index visits (Fig. S3), we identified that the alpha diversity prior to incident *Shigella* infection did not differ between the cases and the controls. (Table S3). In addition, no taxa were significantly associated with case status (*Shigella*-infected versus controls) during this time interval (Fig. S3).

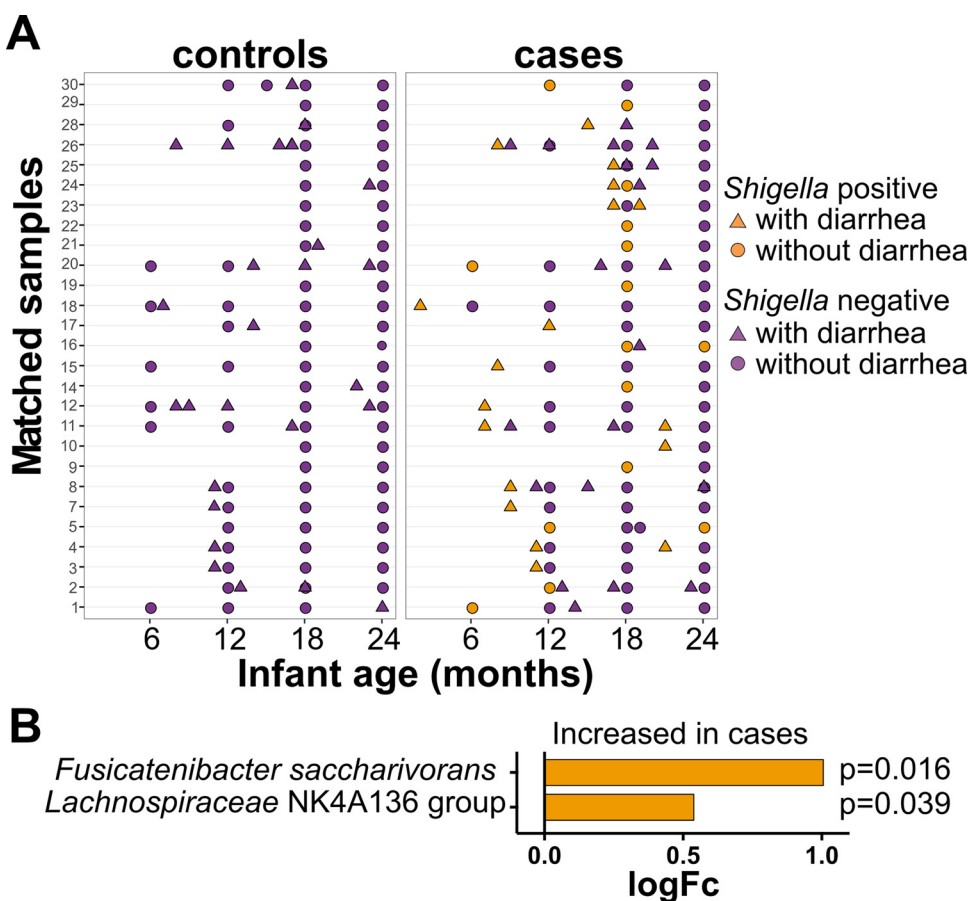

**FIG 4** Taxa differentially abundant in cases after *Shigella* infection. (A) Chart representing samples that were included in the analysis. All samples after the collection of a *Shigella* qPCR positive sample in the case individuals (right column) and their matching controls (left column). Samples in orange represent those that were found to be *Shigella* qPCR positive. (B) Taxa identified to be significantly (adjusted $P < 0.05$) abundant in cases versus controls after *Shigella* infection. The $x$ axis indicates the estimated difference (log$_2$-fold change, logFc) between the abundance of taxa in the cases compared to the controls. These estimates were obtained from logistic regression models. A positive logFc indicates a greater abundance in the cases than in the controls. The adjusted $P$ values were determined using a mixed effects linear regression model controlled for match, sample age, infant ID, diarrhea, and short-term antibiotic use.

To determine the effects of *Shigella* infection on subsequent microbial community composition, we compared the SDI from all samples after the case and control index visits (Fig. 4A). *Shigella* infection did not significantly alter the SDI of cases postinfection compared to their matched controls (Table S3). However, abundances of *Fusicatenibacter saccharivorans* and *Lachnospiraceae* NK4A136 group, both of which are members of the family Lachnospiraceae, were significantly more abundant in the cases than in the controls following *Shigella* infection (Fig. 4B). This was also observed when comparing the relative abundance of both species over time. There was an increase in the relative abundance of both species after the first year, which was significant for *Fusicatenibacter saccharivorans* (Fig. S5A). When comparing the cases versus the controls, there was a trend of greater abundance in the cases versus the controls for both species at months 18 and 24 (Fig. S5B).

**Shigella-associated diarrhea results in increases in taxonomic groups that are distinct from infection with other pathogens.** Multiple studies in children younger than 5 years of age have identified shifts in microbiome composition associated with infectious diarrhea, as measured by the SDI (32, 33, 38). We examined the specific impact of *Shigella*-associated diarrhea on microbiota composition in our cohort in two ways. First, we compared the microbiota of the case individuals who were symptomatic (with diarrhea) with those who were asymptomatic (without diarrhea) before and after *Shigella* infection. The SDI values for the microbiomes of infants who had a

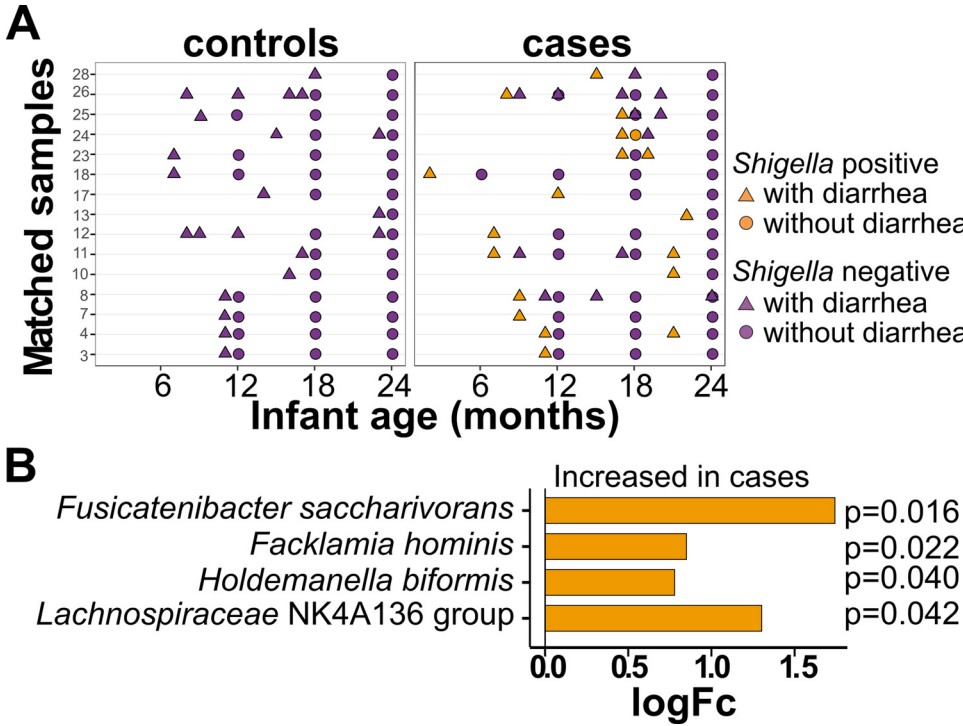

**FIG 5** Taxa differentially abundant in cases after *Shigella*-driven diarrhea versus diarrhea from other causes. (A) Chart representing samples that were included in the analysis. Case samples are those that were collected from case individuals after a *Shigella* qPCR positive and diarrhea positive sample (right column, orange triangles). Control samples are those from matched control individuals after a diarrhea-positive sample was collected within a month of the case sample (left column, purple triangles). (B) Taxa identified to be significantly (adjusted $P < 0.05$) abundant in cases after *Shigella* positive diarrheal infection versus the controls after a diarrhea positive sample was collected. The x axis indicates the estimated difference (log$_2$-fold change, logFc) between the abundance of taxa in the cases compared to the controls. These estimates were obtained from logistic regression models. A positive logFc indicates a greater abundance in the cases than in the controls. The adjusted $P$ values were determined using a mixed effects logistic regression model controlled for infant sex, birth month, sample age, and infant ID.

symptomatic infection were not significantly different from those without diarrhea either before ($P = 0.25$) or after ($P = 0.65$) infection (Table S3). In addition, no taxa were significantly associated with symptomatic *Shigella* infection either before or after infection (Fig. S4).

As diarrhea can result from colonization by multiple different pathogens, we next compared the microbiota from the cases with a diarrhea-positive, *Shigella* qPCR positive sample at the index visit (Fig. 5A, right) with matched control samples (diarrhea-positive and *Shigella* qPCR negative) collected within a month of the case sample (Fig. 5A, left). The alpha diversity did not significantly differ following the *Shigella*-associated diarrheal visits compared to the diarrheal visits not associated with *Shigella* infection ($P = 0.42$) (Table S3). However, *Facklamia hominis* and *Holdemanella biformis*, both of which are members of the Firmicutes phylum, and the two species that had been identified as increased in the cases after *Shigella* infection, namely, *Fusicatenibacter saccharivorans* and *Lachnospiraceae* NK4A136 group, were significantly increased in the cases after *Shigella*-associated diarrhea (Fig. 5B).

## DISCUSSION

The first years of life are a critical period for the colonization, expansion, and maturity of the gastrointestinal microbiome, which in turn is essential for appropriate mucosal immune development (30, 31). Enteric infections during this time may alter the gut microbiome (32, 33) with long-lasting consequences. Here, we examined the dynamics of the infant gastrointestinal microbial communities in *Shigella*-infected and noninfected children over the first 2 years of life. Several large field studies of the etiology of

diarrhea in developing countries have shown that *Shigella* infection peaks between the first and second years of life (3, 43). This was also observed in our Malawi cohort, in which *Shigella* cases progressively increased toward the second year of life (Table 1). Hence, the timing of our microbiome analysis is relevant, as it spans the critical age range when the risk of infection is highest. The analysis of this particular age group was important in attempting to identify perturbations that may be associated with repeated illness and long-lasting health impairments.

Overall, in all children, the gastrointestinal microbiota was dominated by the species *Bifidobacterium longum* at 6 months of age, and this was replaced by the abundance of various *Prevotella* spp. by the time the infants reached the 2-year mark. A previous study of microbiomes of children from Malawi at only 2 time points, 6 and 18 months of age, identified an increase of *Prevotella* that tracked with age, with a corresponding decrease of Bifidobacteriaceae and Enterobacteriaceae (57). *Bifidobacterium* spp. have been demonstrated to be enriched in the gastrointestinal microbiome of breastfed infants (49, 50). Therefore, the decrease of these species that we observed is consistent with that observed in children who are being weaned from breastfeeding.

We found that the alpha diversity increased during the first 2 years of life (Fig. 2A), which was consistent with previous reports (32, 54). Unlike previous studies that reported a decrease in alpha diversity after diarrheal infection (32, 33), we observed in our study that *Shigella* infection alone did not alter the trajectory of alpha diversity in infants over the first 2 years of life (Fig. 2B). This finding suggests that the competition between *Shigella* and the commensals in the gastrointestinal milieu may open a niche, allowing for the expansion of other bacteria to maintain microbiota diversity. Alternatively, the stability in alpha diversity may reflect the dynamic nature of the infant gastrointestinal microbiota that allows for recovery after the clearance of *Shigella*. This may be readily appreciated in a longitudinal, as opposed to a cross-sectional, study. It is also possible that our cohort was relatively healthy and that studies that have shown a decrease in alpha diversity compared healthy children with children who experienced more severe disease (32) than did those in our cohort.

The longitudinal design of our study also allowed us to infer a cause-and-effect relationship between the composition of the microbiota and *Shigella* infection. We had hypothesized that the presence of specific microbial communities could impact (i.e., increase susceptibility and help predict) infection status. Surprisingly, no single species, not even *Escherichia/Shigella*, was identified as significantly associated with susceptibility to *Shigella* infection. The latter is probably due to the inability to separate *Escherichia* species from *Shigella* and the presence of *E. coli* as a constituent of the healthy microbiome (58–60). This observation also suggests that *Shigella* infection is not only dependent on the presence of that organism but is determined by other host-related factors, such as maternal immunity (i.e., antibodies in breast milk) and the individual's own local innate defenses. We and others have identified placentally acquired maternal antibodies to *Shigella* antibodies at birth (16, 17, 41). *Shigella*-specific antibodies have been found in breast milk (61). Host (infant) innate immunity also likely influences infection outcome. Further studies are needed to ascertain the role of maternal and infant immune components in preventing infection in early life.

Recovery from a *Shigella* infection was linked with a distinct alteration in microbiota composition. The species *Lachnospiraceae* NK4A136 group and *Fusicatenibacter saccharivorans* were increased in children who had a *Shigella* infection compared to controls (Fig. 4B), and this was also observed in children with *Shigella*-associated diarrhea (Fig. 5B). In a study that compared the microbiota profiles of obese and lean adults in Spain, the *Lachnospiraceae* NK4A136 group was found to be negatively associated with cardiovascular risk factors, including body fat, LDL, and total cholesterol levels (62). *F. saccharivorans* has been associated with a reduction in intestinal inflammation: it was found to be depleted in patients with active ulcerative colitis, (63). In the same study, the authors showed that the daily administration of heat-inactivated *F. saccharivorans* counteracted colitis symptoms in a mouse model of colitis (63). Both species are part of the Family Lachnospiraceae, whose members are known for their production of beneficial metabolites and fermentation of fiber and plant carbohydrate to short-chain fatty acids (SCFA), specifically butyrate (reviewed in reference [64]).

Multiple *in vivo* and *in vitro* studies have demonstrated butyrate to display multiple roles in the gut epithelium, including colonocyte proliferation and differentiation, and it has been proposed to have anti-inflammatory effects, all of which point to its capacity to promote gut health (reviewed in reference [65]). Species in the Family Lachnospiraceae have been associated with recovery from other enteric infections, including *Vibrio cholerae* in 2- and 3-year-old children (35) and traveler's diarrhea (66) in adults. Two other taxa, *Facklamia hominis* and *Holdemanella biformis*, both part of the Firmicutes phylum, were also increased after *Shigella*-associated diarrheal infection. *F. hominis* is an uncommon pathogen that has been reported in a few cases of human infections (67). *H. biformis* was found to be reduced in the microbiota of patients with colorectal adenomas, and it also produces SCFAs that reduce tumor cell proliferation (68). In addition, *H. biformis* was shown to produce long-chain fatty acids (LCFAs) that reduced inflammation in a dextran sulfate sodium (DSS)-induced mouse colitis model (69). *Shigella* infection produces an extensive inflammatory local response (70). It is likely that an increase in species that reduce inflammation and promote broad gastrointestinal health improvements represents an attempt to stabilize the commensal repertoire following infection.

Several ways in which members of the family Lachnospiraceae would facilitate the recovery from *Shigella* infection can be envisioned. Another member of this family, *Ruminococcus obeum* (reclassified as *Blautia obeum* [71]), was correlated with gut microbiota recovery from cholera in Bangladeshi infants, and it was shown to restrict colonization by *V. cholerae* in a mouse model (36). Notably, *B. obeum* was increased among *Shigella* cases at the index visit in our study (Fig. S2), though this increase was not statistically significant after controlling for multiple variables. The "recovery species" may also interact with other bacteria, their metabolites, or the host immune system (29, 72). A study of 6- to 24-month-old malnourished children in Gambia implicated species belonging to Lachnospiraceae in observed shifts from acute malnutrition following nutritional interventions (26). While the infants' diets were not recorded in our study, a similar recovery effect may be plausible in our study. Whether the changes we observed are a natural progression of the microbiota in response to *Shigella* infection or are caused by environmental factors (e.g., antibiotics, oral rehydration, diet) warrants investigation in future studies.

An unexpected finding from our study was the identification of the *Bifidobacterium kashiwanohense* taxa, which not only was one of the most abundant but also displayed the opposite trend expected of other *Bifidobacterium* spp., increasing significantly over the first 2 years of life (Fig. 3). To our knowledge, this species has not been reported as part of the developing early life gastrointestinal microbiome, although it had been reportedly cultured from feces from a healthy Japanese infant (73). *B. kashiwanohense* has also been transiently identified in formula-fed (compared to breastfed) infants in Thailand (74). Like others in the genus, isolates of *B. kashiwanohense* have been shown to utilize the human milk oligosaccharide component, fucosyllactose (75, 76), yet the significance of this process within the gastrointestinal environment is unknown. Interestingly, in a study of anemic (compared to healthy) 6-month-old Kenyan infants, *B. kashiwanohense* was enriched in the anemic infants (77), likely due to its ability to bind to iron. *B. kashiwanohense*'s iron-sequestration function was also associated with the inhibition of growth and the adhesion of the enteropathogens *Salmonella* Typhimurium and *Enterohemorrhagic Escherichia coli* (EHEC) *in vitro* (78). Further studies incorporating diverse populations would be needed to identify the prevalence of *B. kashiwanohense* in early life gastrointestinal microbiota and to elucidate its role in relation to microbiome composition, infant nutrition, and its potential to control enteric infections.

Our study is the first involving a longitudinal analysis of the microbiota associated with *Shigella* infection in a well-characterized clinical cohort from birth up to 2 years (the age range of the highest risk of infection) that compares cases to matched controls. One limitation is that the 16S rRNA gene amplicon only amplifies bacterial species and therefore did not allow us to analyze the effects of other microbial species (e.g., viruses, fungi, or protozoa) that could affect the incidence and outcomes of shigellosis. Additionally, we could not confirm that *Shigella* was the only etiological agent, as children could be colonized by

multiple diarrheal pathogens (40, 79), all of which could have influenced the outcomes. The duration between the initiation of diarrheal symptoms and sample collection was also not recorded, which may have affected our ability to discern significant microbiome changes. This variable will need to be accounted for in future studies, possibly by increasing sampling density and metadata collection. Nutritional information, including data on breastfeeding patterns, weaning, and infant diets, were not captured in our study, and we were unable to infer their effects on the outcomes measured. Several *Shigella* control human infection models are being pursued with adult volunteers. Such studies offer an opportunity to interrogate *Shigella*-microbiome associations in a setting of controlled dosing, active monitoring (with extensive sampling), and reduced variability.

In summary, in this study, we have characterized the infant gastrointestinal microbiota in children from birth to 2 years of age living in Malawi, and the information generated adds to the limited studies of the gastrointestinal microbiomes of African populations (80). We also demonstrated that *Shigella* infection did not profoundly impact overall species diversity but led to the expansion of species known to improve gastrointestinal health and drive the microbiota back to homeostasis. The exploration of the impact and interaction of these species with *Shigella* during and after infection may be warranted to identify protective elements and therapeutic targets. Therefore, our work provides a foundation for the interrogation of the dynamic nature of the pediatric gastrointestinal microbiome in health and disease.

## MATERIALS AND METHODS

**Study participants and sample collection.** The study participants were a cohort of children from birth to 2 years of age enrolled in a longitudinal malaria surveillance study in Malawi (41) who were also monitored for diarrheal disease. Mothers and infants were recruited between January and November of 2016 at Mfera Health Clinic in Chikwawa, Malawi, and they were followed for 2 years. Healthy pregnant women who were HIV seronegative were enrolled either at their prenatal clinic visit or during their stay for delivery. The infants were enrolled at birth, and their ages, sexes, and baseline health information (e.g., birth weights and lengths) were recorded. Infant health information, including diarrhea status (as defined by the health care provider at the time of the sample) and any antibiotics prescribed, was also obtained at each subsequent visit to the health clinic. All but two of the infants (one case and one control) were prescribed antibiotics (amoxicillin, co-trimoxazole, metronidazole, or erythromycin) at least once during the 2 years. Short-term antibiotic treatment, defined as any antibiotic given within 2 weeks before sample collection, was therefore included as a covariate in our subsequent analyses. While breastfeeding was common up to 24 months of age, data on the duration of exclusive breastfeeding, weaning, the introduction of solid food, and the infants' diets were not available. Therefore, these factors were not included as covariates in our analysis.

Rectal swab samples were collected every 6 months at each scheduled well-child visit as well as at every time that the infant presented at the clinic with diarrhea to diagnose *Shigella* infection. Samples were immediately placed in 350 $\mu$L of DNA/RNA Shield (Zymo Research, Irvine, CA), frozen, and stored at −80°C. Only infants with a complete set of samples obtained at scheduled clinic visits at 6, 12, 18, and 24 months were studied. Cases were defined as children with at least one *Shigella* qPCR positive sample during the 2-year period. Controls were selected as children without a *Shigella* qPCR positive sample during the 2-year period, and they were matched for sex and age (within 1 month of birth where possible) (Fig. 1A).

This study was approved by the Institutional Review Board of the University of Maryland School of Medicine (UMB IRB No: HP-00087456) and the College of Medicine Research and Ethics Committee (COMREC) at the College of Medicine in Malawi (COMREC Ref. No: P.01/16/1859). All of the participating mothers provided written informed consent for themselves and for their infants.

**Rectal swab genomic DNA extraction and *Shigella* qPCR analysis.** DNA was isolated from 100 $\mu$L aliquots of the rectal swab samples using the QIAamp DNA Stool Extraction Kit (Qiagen, Valencia, CA, USA), according to the manufacturer's protocol. Extracted DNA was precipitated with ethanol in a final elution volume of 200 $\mu$L. Each DNA sample was tested for *Shigella* by qPCR using the previously described SYBR green-based fluorescent dye method to detect the *ipaH* gene (81). The primers for *ipaH* were originally created by Vu et al. (82). Quantitation cycle (Cq) values of <35 detected in duplicate wells were required to consider a sample qPCR positive. Controls for the qPCR assay included extraction-negative, qPCR negative, and a known qPCR positive control. While this method cannot distinguish between *Shigella* and enteroinvasive *Escherichia coli* (EIEC), EIEC prevalence is minimal compared with *Shigella* in similar geographic settings (43, 83). Therefore, all of the *ipaH* positive samples were assumed to be *Shigella* (79).

**Rectal swab genomic DNA extraction, amplicon sequencing, and taxonomic assignment.** The remainder of the frozen rectal swab samples were transported to the University of Maryland School of Medicine in Baltimore, and sequencing and sequence processing were performed at the Institute for Genome Sciences Microbiome Service Laboratory (https://msl.igs.umaryland.edu/). Total nucleic acids were

extracted from the remaining (~200 $\mu$L) aliquot of the rectal swab sample using the Qiagen Microbiome RNA/DNA isolation kit (Qiagen, Hilden, Germany) using a semiautomated protocol for the Hamilton Star platform (Hamilton Company, Reno, NV), following the manufacturer's protocol (84). Cells were lysed via physical disruption on the TissueLyser (Qiagen, Hilden, Germany) at 20 Hz for 20 min, and the final elution volume was 110 $\mu$L. 10 negative controls of water were extracted in the same manner (extraction negatives). For the rectal swab samples, as well as for five positive controls (Zymobiomics Microbial Community Standard) and five negative controls (PCR negatives), amplicon sequencing of the 16S rRNA gene V3-V4 variable regions was performed using the 2-Step PCR method and sequences processed as described in Holm et al. (85). Briefly, sequences were demultiplexed using the dual-barcode strategy, a mapping file linking the barcode to the samples, and split_libraries.py, a QIIME-dependent script (86). The resulting forward and reverse fastq files were split by sample using the QIIME-dependent script split_sequence_file_on_sample_ids.py, and the primer sequences were removed using TagCleaner (version 0.16) (87). Further processing followed the DADA2 Workflow for Big Data and dada2 (v. 1.5.2) (https://benjjneb.github.io/dada2/bigdata.html) (88). The forward and reverse reads were each trimmed using lengths of 255 and 225 bp, respectively, filtered to contain no ambiguous bases with a minimum quality score of two, and required to contain less than two expected errors based on their quality scores. The relationship between quality scores and error rates was estimated on the combined sequencing runs to reduce batch effects arising from run-to-run variability. The reads were assembled into amplicon sequence variants (ASV) and chimeras for the combined runs removed as per the dada2 protocol. Taxonomy was assigned using the RDP Classifier (89) and the SILVA database (v138) (90). For each amplicon sequence variant, we applied the lowest (finest) level of taxonomy possible and then combined the counts of the ASVs with the same taxonomic assignment. Thus, some ASVs were assigned at the level of species, of genus, etc.

The maximum number of reads in the PCR negative controls was 1,233. The genus *Sphingomonas* was observed in most samples as well as in the extraction negative and the PCR negative controls. Therefore, it was removed from all samples prior to the downstream analyses. Taxa identified as d_Bacteria were also removed, as these contained counts for unidentified reads. Additionally, taxa observed in <5% of samples study-wide were removed ($n = 1,120$ taxa). The mean number of reads per sample for taxa prevalent in >5% of the samples remained close to the original number (41201.36). Samples with fewer than 1,000 reads were culled ($n = 10$ samples). Following the filtering steps, the 16S rRNA gene amplicon libraries yielded an average of 23,720 reads per sample with a total of 8,729,027 reads (Data Set S4). A total of 325 taxa were identified after quality filtering. All data are provided in Data Set S5.

**Statistical analyses.** The length for age *z* scores (LAZ) were calculated based on the World Health Organization child growth standards (91) and using the zscorer package (v0.3.1) in R. Population characteristics (birthweight, LAZ, maternal age at birth, and gestational age at birth) between the cases and the controls were compared using unpaired *t* tests in GraphPad Prism v9 (San Diego, CA). *P* values of <0.05 were considered significant.

The alpha diversity was calculated using the Shannon diversity index (SDI) for each sample via the diversity function from the vegan package (v. 2.5-7) in R (92). Correlations between SDI and infant age were determined via linear regression (lm function in R) (93). Here, infant age was the predictor variable, and SDI was the response. To test for an association between case status and changes in alpha diversity over time (age), the data were fit to a mixed effects linear regression model which accounted for match and for an interaction term between age and case status. *P* values of <0.05 were considered to be indicative of a statistically significant result.

Index visits were defined as the first visit at which a *Shigella* qPCR positive test was observed. The first visit was used for individuals who had more than one *Shigella* qPCR-positive sample. For the matched controls, the index visit was the sample from the same age (within 1 month). Alpha diversity at index visits was compared using a two-way analysis of variance (ANOVA) (Type III) analysis in which SDI was the response variable, case status was the predictor, and the following covariates were adjusted for: infant age, infant sex, month of sample collection, diarrhea status, and short-term antibiotic use. An adjusted *P* value of <0.05 was considered to be indicative of a statistically significant result.

The effect of case status on alpha diversity prior to incident infection was compared using a generalized logistic regression model where the response was case status and the predictor was SDI. Comparisons after infection were made using a mixed effects linear regression model in which the response was SDI and the predictor was case status. Covariates included in these models were matched sample (as per Fig. 1A), sample age, infant ID (to account for multiple samples collected from the same individual), diarrhea status, and short-term antibiotic use. An adjusted *P* value of <0.05 was considered to be indicative of a statistically significant result. Relationships were visualized using ggplot2 (v3.3.5) (94).

Regarding differential abundance, for each of the top 10 taxa, the mean relative abundances at months 6, 12, 18, and 24 were compared using a one-way ANOVA. The mean relative abundances at months 6, 12, 18, and 24 between the cases and the controls were compared using unpaired *t* tests. *P* values of <0.05 were considered to be indicative of a statistically significant result. The statistical analysis was conducted using GraphPad Prism v9.

To identify associations between taxa abundance and case status before and after *Shigella* infection, mixed effects logistic or linear regression models were used, respectively, adjusting for the following covariates: matched sample (as per Fig. 1A), sample age, infant ID, diarrhea status, and short-term antibiotic use. An adjusted *P* value of <0.05 was considered to be indicative of a statistically significant result.

To compare taxa that were differentially abundant before or after *Shigella* with or without diarrhea infection, a mixed effects logistic regression model was used, accounting for infant sex, sample age, birth month, infant ID, and short-term antibiotic use. A similar model was used to compare taxa after

obtaining diarrheal samples with *Shigella* infections as opposed to infections of other pathogens. An adjusted *P* value of <0.05 was considered to be indicative of a statistically significant result.

Count data were normalized for differences in coverage using the "poscounts" estimator (deals with a gene with some zeros by calculating a modified geometric mean by taking the *n*th root of the product of the nonzero counts) in the DESeq2 package in R (95). A local regression of log dispersions was fit over the log base mean. *P* values were obtained using Wald's test (uses the estimated standard error of the log$_2$-fold change between conditions to test whether it is equal to zero). The *P* values for the multiple comparisons were adjusted using the false discovery rate (FDR). An adjusted *P* value of <0.05 was considered to be indicative of a statistically significant result.

**Data availability.** All of the raw sequencing data were deposited into NCBI SRA under BioProject ID PRJNA834726.

## SUPPLEMENTAL MATERIAL

Supplemental material is available online only.

**DATA SET S4**, XLSX file, 0.04 MB.
**DATA SET S5**, XLSX file, 0.8 MB.
**FIG S1**, TIF file, 1.2 MB.
**FIG S2**, TIF file, 1.1 MB.
**FIG S3**, TIF file, 0.4 MB.
**FIG S4**, TIF file, 0.8 MB.
**FIG S5**, TIF file, 0.3 MB.
**TABLE S1**, DOCX file, 0.01 MB.
**TABLE S2**, DOCX file, 0.01 MB.
**TABLE S3**, DOCX file, 0.01 MB.

## ACKNOWLEDGMENTS

This work was supported by the National Institutes of Health awards R01AI117734 and R01AI125841, as well as by Research Supplement to Promote Diversity in Health-Related Research awards 3R01AI117734-04S1 to M.F.P., U19AI110820 to D.A.R., and T32DK067872 to E.N. The sequencing costs were subsidized by the University of Maryland, Baltimore, Institute for Clinical & Translational Research and by the National Center for Advancing Translational Sciences (NCATS) Clinical Translational Science Award (CTSA), grant number 1UL1TR003098.

We thank Mike Humphrys and Justin Grant at the Microbiome Service Laboratory for sequencing support. We are also grateful to the support team at the Mfera Health Center in Malawi and to the mothers, infants, and families who volunteered to participate. We thank members of the Laufer and Pasetti labs for providing technical and logistic expertise during the sample collection process and for discussion.

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
