## [Reviewer comments · mSystems]

Dynamics of the gut microbiome in *Shigella*-infected children during the first two years of life

Esther Ndungo, Johanna Holm, Syze Gama, Andrea Buchwald, Sharon Tennant, Miriam Laufer, Marcela Pasetti, and David Rasko

Corresponding Author(s): David Rasko, University of Maryland School of Medicine

Review Timeline:

Submission Date:	May 13, 2022
Editorial Decision:	June 13, 2022
Revision Received:	July 28, 2022
Accepted:	August 23, 2022

Editor: Ryan McClure

Reviewer(s): The reviewers have opted to remain anonymous.

Transaction Report:

DOI: <https://doi.org/10.1128/msystems.00442-22>

June 13, 2022

Dr. David Rasko
University of Maryland School of Medicine
Institute for Genome Sciences
670 W. Baltimore Street, Room 2104
Baltimore, MD 21201

Re: mSystems00442-22 (Dynamics of the gut microbiome in *Shigella*-infected children during the first two years of life)

Dear Dr. David Rasko:

Thank you for submitting your manuscript to mSystems. We have completed our review and I am pleased to inform you that, in principle, we expect to accept it for publication in mSystems. However, acceptance will not be final until you have adequately addressed the reviewer comments. In addition, you will need to include a "Data Availability" paragraph as detailed in mSystems author information documents.

Preparing Revision Guidelines

Sincerely,

Ryan McClure

Editor, mSystems

Journals Department
American Society for Microbiology

Reviewer comments:

Reviewer #1 (Comments for the Author):

This work presents interesting results of microbial population shifts in the infant gut microbiota using the amplicon sequencing approach. The study undertakes a well-designed approach inclusive of all required controls and a nice downstream data analysis scheme.

Minor comments:

1. Line 135: 'we focused on the top 10 identified taxa based on relative abundance for all individuals in the cohort'

The top 10 taxa in terms of abundance have been included, the 'others' category hence could be inclusive of the remaining identified and unidentified populations. Was any particular trend observed in the case of unidentified populations? Also, it would be nice to see the list expanded to the top 15-20 or more and get insight into other diverse microbes present.

2. Line 141: 'well as Escherichia/Shigella, (16S rRNA sequencing does not distinguish these separately)' and predict' and line 254: 'the inability to separate Escherichia species from Shigella, and the presence of E. coli as a constituent of the healthy microbiome (61-63)'

This study focuses on the specific impact of Shigella-associated diarrhea. As already identified from other reports E. coli often occurs in healthy gut microbiomes and hence a distinction between the two genera would be valuable. Could the authors subject a subset of positive samples to qPCR using Shigella specific marker gene (e.g chromosomal IpaH)?

3. Line 160-161: "While SDI was significantly associated with infant age (recent antibiotic use ($p=0.017$) (Table S2), alpha diversity did not differ significantly by Shigella infection status ($p=0.52$) (Table S3)".

The use of antibiotics in infants does impact the gut microbiome (Gibson et al. 2015; Korpela et al 2020). However, the diversity index does not vary in response to the use of antibiotics in this study. Please comment.

Reviewer #2 (Comments for the Author):

Shigella infections pose a serious health risk, especially for children under the age of five. The role of the infant microbiome before and after a Shigella infection is unclear. This study takes a longitudinal approach to address this important issue. Overall, this paper is well-written, and the research is well carried out. It is quite surprising that Shigella infections hardly seem to disrupt the evolution of the infant microbiome.

Line 176: Since Fusicatenibacter saccharivorans and Lachnospiraceae NK4A136 were the only two whose abundance significantly changed following Shigella infection, it would be relevant to include these two in Fig 3B. Do their abundances naturally change with increasing infant age?

Discussion: While not entirely necessary, this section would benefit from discussing the primary topic of this paper, i.e., Shigella infections, prior to B. kashiwanohense.

Line 204: Are any of the taxa presented within this paper relevant for the development of the mucosal barrier? Was there any evidence of Shigella disrupting taxa that have been implicated in altering mucosal immune development? While minor, these sentences distract the reader from the focus of this paper and could be better rooted within the data or removed.

Line 227: "B. kashiwanohense has also been found to be transiently expressed..." Transiently found is a much more accurate term as genes are expressed.

Lines 232-236: Were B. kashiwanohense levels correlated with a Shigella infection? Or just an increasing age? Otherwise, it seems suggestive to discuss how B. kashiwanohense may control infections (i.e., Salmonella and EHEC).

Line 246: Did the infants experience symptoms for a period of a couple days or weeks? If the infection was acute, could it be that this disturbance was not prolonged enough to significantly alter microbial diversity? Also, could it be that any alterations in microbial diversity would be most significant immediately following an infection as opposed to months after (i.e., did microbial

diversity equilibrate over time)? This could be a discussion point.

Line 291: What is the relevance of *Ruminococcus obeum*? This is not mentioned anywhere in the Results section. It would be better to root the envisioned *Shigella* recovery using data gathered in this paper.

Shigella infections pose a serious health risk, especially for children under the age of five. The role of the infant microbiome before and after a *Shigella* infection is unclear. This study takes a longitudinal approach to address this important issue. Overall, this paper is well-written, and the research is well carried out. It is quite surprising that *Shigella* infections hardly seem to disrupt the evolution of the infant microbiome.

Line 176: Since *Fusicatenibacter saccharivorans* and *Lachnospiraceae* NK4A136 were the only two whose abundance significantly changed following *Shigella* infection, it would be relevant to include these two in Fig 3B. Do their abundances naturally change with increasing infant age?

Discussion: While not entirely necessary, this section would benefit from discussing the primary topic of this paper, i.e., *Shigella* infections, prior to *B. kashiwanohense*.

Line 204: Are any of the taxa presented within this paper relevant for the development of the mucosal barrier? Was there any evidence of *Shigella* disrupting taxa that have been implicated in altering mucosal immune development? While minor, these sentences distract the reader from the focus of this paper and could be better rooted within the data or removed.

Line 227: "*B. kashiwanohense* has also been found to be transiently expressed..." Transiently found is a much more accurate term as genes are expressed.

Lines 232-236: Were *B. kashiwanohense* levels correlated with a *Shigella* infection? Or just an increasing age? Otherwise, it seems suggestive to discuss how *B. kashiwanohense* may control infections (i.e., *Salmonella* and EHEC).

Line 246: Did the infants experience symptoms for a period of a couple days or weeks? If the infection was acute, could it be that this disturbance was not prolonged enough to significantly alter microbial diversity? Also, could it be that any alterations in microbial diversity would be most significant immediately following an infection as opposed to months after (i.e., did microbial diversity equilibrate over time)? This could be a discussion point.

Line 291: What is the relevance of *Ruminococcus obeum*? This is not mentioned anywhere in the Results section. It would be better to root the envisioned *Shigella* recovery using data gathered in this paper.

Response to reviewers for mSystems00442-22

We thank the reviewers and the editor for their assessment of our work and for the positive comments and helpful critiques. We have addressed the comments in point-by-point responses below. The responses are colored in blue below.

Reviewer comments:

Reviewer #1 (Comments for the Author):

This work presents interesting results of microbial population shifts in the infant gut microbiota using the amplicon sequencing approach. The study undertakes a well-designed approach inclusive of all required controls and a nice downstream data analysis scheme.

Response: We thank the reviewer for the positive comments.

Minor comments:

1. Line 135: 'we focused on the top 10 identified taxa based on relative abundance for all individuals in the cohort'

The top 10 taxa in terms of abundance have been included, the 'others' category hence could be inclusive of the remaining identified and unidentified populations. Was any particular trend observed in the case of unidentified populations? Also, it would be nice to see the list expanded to the top 15-20 or more and get insight into other diverse microbes present.

We agree with the reviewer that a more complete data set is always desirable. As indicated in the Methods, analysis was performed on 325 total taxa after filtering steps, which included removing taxa found in the extraction and PCR negative controls, taxa observed in < 5% of samples study-wide, and those classified as "d_Bacteria," which were unidentifiable reads. Pursuant to the reviewer's comment, we have opted to provide all the data in the Supplemental material, including the relative abundance of the 325 taxa for all samples, so that interested researchers have the ability to perform further analysis themselves as desired.

We prefer to maintain the figure showing the top 10 taxa as they represent between 40-80% of the taxa present in children for all ages (Fig. 3A); including more taxa to this figure will make it more complex and difficult to understand.

Modifications made to the manuscript: We have included this dataset as Supplemental Table S5, and added a line in the Methods for clarity:

Line 410: Taxa identified as d_Bacteria were also removed as these contained counts for unidentified reads.

2. Line 141: 'well as Escherichia/Shigella, (16S rRNA sequencing does not distinguish these separately)' and predict' and line 254: 'the inability to separate Escherichia species from Shigella, and the presence of E. coli as a constituent of the healthy microbiome (61-63)'.

This study focuses on the specific impact of Shigella-associated diarrhea. As already identified from other reports E. coli often occurs in healthy gut microbiomes and hence a distinction between the two genera would be valuable. Could the authors subject a subset of positive samples to qPCR using Shigella specific marker gene (e.g chromosomal IpaH)?

We appreciate the comment by the reviewer, and this is an excellent suggestion. One of the caveats of completing this exercise would be to have a reliable marker to distinguish both *Shigella* and *E. coli*. As the reviewer points out, the *ipaH* gene would be a single copy gene marker to identify *Shigella*; however, a common and conserved single copy *E. coli* marker exclusive to all *E. coli* that will universally identify this species and not *Shigella* is lacking. The qPCR data included in Figure S1 is focused on the *ipaH* gene and thus we can identify the samples with *Shigella*. Unfortunately, the relative signal from the *Shigella* versus the *E. coli* in the 16S rRNA data is not currently attainable. The separation of these two genera is an unmet research gap as a definitive method does not exist.

3. Line 160-161: "While SDI was significantly associated with infant age (recent antibiotic use ($p=0.017$) (Table S2), alpha diversity did not differ significantly by *Shigella* infection status ($p=0.52$) (Table S3)".

The use of antibiotics in infants does impact the gut microbiome (Gibson et al. 2015; Korpela et al 2020). However, the diversity index does not vary in response to the use of antibiotics in this study. Please comment.

We thank the reviewer for pointing out these additional publications addressing the impact of antibiotic use and the gut microbiota in infants. The sentence in our manuscript mentioned by the reviewer may not have clearly indicated that the microbiome was indeed altered with antibiotic treatment in agreement with the referenced studies.

In our study, we identified that the alpha diversity was significantly associated with recent antibiotic use ($p=0.017$, Table S2) and for this reason, we included antibiotic use as a covariate in our subsequent analyses (Line 350).

Modifications made to the manuscript: We have modified the sentence for clarity and added the references suggested by the reviewer:

Line 160: After adjusting for variables that could affect alpha diversity, including infant age (32, 33, 38), diarrhea (32, 54), and antibiotic use (55, 56), we determined that the SDI did not differ significantly by *Shigella* infection status ($p=0.52$) (Table S2). Notably, alpha diversity was significantly associated with infant age ($p<0.001$), diarrhea ($p=0.012$), and recent antibiotic use ($p=0.017$) (Table S2).

Reviewer #2 (Comments for the Author):

Shigella infections pose a serious health risk, especially for children under the age of five. The role of the infant microbiome before and after a *Shigella* infection is unclear. This study takes a longitudinal approach to address this important issue. Overall, this paper is well-written, and the research is well carried out. It is quite surprising that *Shigella* infections hardly seem to disrupt the evolution of the infant microbiome.

We thank the reviewer for the positive assessment and for recognizing the novelty of the study.

Line 176: Since *Fusicatenibacter saccharivorans* and *Lachnospiraceae* NK4A136 were the only two whose abundance significantly changed following *Shigella* infection, it would be relevant to include these two in Fig 3B. Do their abundances naturally change with increasing infant age?

We thank the reviewer for this suggestion. To address this comment, we plotted the distribution of these two taxa, *Fusicatenibacter saccharivorans* and *Lachnospiraceae* NK4A136, over time,

and comparing cases and controls (Supplementary Fig. S5A and B). We identified a significant increase in the relative abundance of *Fusicatenibacter saccharivorans*, especially after the first year. There was a similar trend for *Lachnospiraceae* NK4A136, but this was not significant. Further, there was a trend towards a greater relative abundance of both species in cases compared to controls, especially at 18 and 24 months of age. This suggests that the increase in abundance observed after the first year reflects our observation of an increased abundance in these taxa in cases following infection.

Modifications made to the manuscript: We have added these 4 new graphs as Supplemental Fig. S5, and a sentence has been added to the Results:

Line 181: This was also observed when comparing the relative abundance of both species over time. There was an increase in the relative abundance of both species after the first year, which was significant for *Fusicatenibacter saccharivorans* (Fig. S5A). When comparing the cases versus control, there was a trend of higher abundance in the cases versus the controls for both species at month 18 and 24 (Fig. S5B).

Discussion: While not entirely necessary, this section would benefit from discussing the primary topic of this paper, i.e., *Shigella* infections, prior to *B. kashiwanohense*.

We appreciate the comment by the reviewer. As the reviewer correctly points out, our observation of *B. kashiwanohense* as one of the top 10 taxa in our cohort is important, but is not the primary focus of this manuscript, which was to analyze the impact of *Shigella* infection on the microbiota in infants.

Modifications made to the manuscript: We have modified the text in the Discussion as suggested by the reviewer. We have moved the discussion on the identification of *B. kashiwanohense* after the discussion of the impact of *Shigella* infection on the microbiota, beginning on Line 295.

Line 204: Are any of the taxa presented within this paper relevant for the development of the mucosal barrier? Was there any evidence of *Shigella* disrupting taxa that have been implicated in altering mucosal immune development? While minor, these sentences distract the reader from the focus of this paper and could be better rooted within the data or removed.

The comment from the reviewer is appropriate and well received.

Modifications made to the manuscript: We have modified the text in the Discussion to reduce the emphasis on mucosal immune development as suggested by the reviewer:

Line 209: The first years of life are a critical period for the colonization, expansion, and maturity of the gastrointestinal microbiome, which in turn is essential for appropriate mucosal immune development (30, 31). Enteric infections during this time may alter the gut microbiome (32, 33) with long lasting consequences.

Line 227: "*B. kashiwanohense* has also been found to be transiently expressed..." Transiently found is a much more accurate term as genes are expressed.

Modifications made to the manuscript: This line has been modified as suggested by the reviewer:

Line 300: *B. kashiwanohense* has also been transiently identified in formula-fed (as compared to breastfed) infants in Thailand (74).

Lines 232-236: Were *B. kashiwanohense* levels correlated with a *Shigella* infection? Or just an increasing age? Otherwise, it seems suggestive to discuss how *B. kashiwanohense* may control infections (i.e., *Salmonella* and EHEC).

We did not observe a correlation between *B. kashiwanohense* prevalence and *Shigella* infection. Little is known about the prevalence or role of this species in the gut microbiota; we included the association between *B. kashiwanohense* and the control of EHEC and *Salmonella in vitro* to highlight how much is unknown, including the mechanism of infection control.

Modifications made to the manuscript: We have modified the text in the Discussion to indicate this point more clearly.

Line 308: Further studies in diverse populations would be needed to identify the prevalence of *B. kashiwanohense* in early life gastrointestinal microbiota and elucidate its role in relation to microbiome composition, infant nutrition, and potential to control enteric infections.

Line 246: Did the infants experience symptoms for a period of a couple days or weeks? If the infection was acute, could it be that this disturbance was not prolonged enough to significantly alter microbial diversity? Also, could it be that any alterations in microbial diversity would be most significant immediately following an infection as opposed to months after (i.e., did microbial diversity equilibrate over time)? This could be a discussion point.

The reviewer makes a good point. We used qPCR to detect *Shigella* infection; given the sensitivity of this assay, we were likely to detect the presence of the organism, whether there was an active infection, or if the infection was cleared. As described in the Methods, subjects were defined as having diarrhea at the time of sampling as defined by the health provider. Other *Shigella*-related symptoms or duration of diarrheal disease were not captured in the dataset, and we therefore cannot evaluate the impact of clinical parameters on microbial diversity. The reviewer raises an important point that will be explored in future studies.

Modifications made to the manuscript: We have added text to the Discussion:

Line 319: The duration between initiation of diarrheal symptoms and sample collection was also not recorded, which may have affected our ability to discern significant microbiome changes. This variable will need to be accounted for in future studies by possible increased sampling density and metadata collection.

Line 291: What is the relevance of *Ruminococcus obeum*? This is not mentioned anywhere in the Results section. It would be better to root the envisioned *Shigella* recovery using data gathered in this paper.

We proposed that *Fusicatenibacter saccharivorans* and *Lachnospiraceae* NK4A136, members of the Family Lachnospiraceae, are increased after *Shigella* infection because they play a role in recovery. We therefore used *Ruminococcus obeum*, another member of the Lachnospiraceae family, as an example of the role of these species in providing beneficial effects against infection.

Modifications made to the manuscript: We have modified the text for clarity:

Line 282: Several ways in which members of the Family Lachnospiraceae would facilitate the recovery from *Shigella* infection could be envisioned. Another member of this family, *Ruminococcus obeum* (reclassified as *Blautia obeum* (71)), was correlated with gut microbiota recovery from cholera in Bangladeshi infants, and was shown to restrict colonization by *V. cholerae* in a mouse model (36).

August 23, 2022

Dr. David Rasko
University of Maryland School of Medicine
Institute for Genome Sciences
670 W. Baltimore Street, Room 2104
Baltimore, MD 21201

Re: mSystems00442-22R1 (Dynamics of the gut microbiome in *Shigella*-infected children during the first two years of life)

Dear Dr. David Rasko:

Your manuscript has been accepted, and I am forwarding it to the ASM Journals Department for publication. For your reference, ASM Journals' address is given below. Before it can be scheduled for publication, your manuscript will be checked by the mSystems production staff to make sure that all elements meet the technical requirements for publication. They will contact you if anything needs to be revised before copyediting and production can begin. Otherwise, you will be notified when your proofs are ready to be viewed.

Publication Fees:

If you would like to submit a potential Featured Image, please email a file and a short legend to mssystems@asmusa.org. Please note that we can only consider images that (i) the authors created or own and (ii) have not been previously published. By submitting, you agree that the image can be used under the same terms as the published article. File requirements: square dimensions (4" x 4"), 300 dpi resolution, RGB colorspace, TIF file format.

We recognize that the video files can become quite large, and so to avoid quality loss ASM suggests sending the video file via <https://www.wetransfer.com/>. When you have a final version of the video and the still ready to share, please send it to mSystems staff at mssystems@asmusa.org.

Sincerely,

Ryan McClure
Editor, mSystems

Journals Department
Fig. S5: Accept
Table S5: Accept
Fig. S2: Accept
Table S3: Accept
Fig. S1: Accept
Fig. S4: Accept
Table S1: Accept
Fig. S3: Accept
Table S4: Accept
Table S2: Accept